# Perspectives of multidisciplinary healthcare providers in elderly daycare centres: Challenges, opportunities and impacts on geriatric care in Chiangrai Municipality, Thailand

**Paralee Opasanunt[¤], Panitsara Leekuan[ID][¤]***

School of Nursing, University of Phayao, Phayao, Thailand

¤ Current address: School of Nursing, University of Phayao, Phayao, Thailand
* panitsara.le@up.ac.th

**Data availability statement:** All relevant data are within the manuscript and its Supporting information files.

## Abstract

The global surge in the ageing population has intensified pressure on healthcare delivery systems, leading to a heightened demand for elderly daycare services. This demographic shift necessitates innovative approaches to meet the complex needs of older adults, including expanding community-based care options. Elderly daycare centers play a crucial role in comprehensive geriatric care by offering services. Understanding how healthcare providers experience their work in Elderly Daycare Centres is essential for improving both service quality and staff preservation. This study seeks the experiences of these providers to understanding of meanings healthcare professionals assign to challenges they face, the successful practices they employ, and highlight opportunities for enhancement within geriatric care.

Interpretative phenomenology influenced by Heidegger's philosophy in qualitative research was conducted. Thirteen semi-structured interviews and narrative accounts with multidisciplinary healthcare providers from four Elderly Daycare Centres were collected and analysed, and interpretative phenomenological analysis (IPA). Multidisciplinary healthcare providers discovered four key aspects of their experience while delivering services in the first two programs (the Muscle Strengthening Program and the Nutrition Program for individuals with nutritional deficiencies): *Opportunities in policy implementation, Challenging experience, Enhancing organisational effectiveness and engaging support, and Reflections on Program Outcomes and Impacts*. All findings mirrored the value and impact of providing healthcare services in Elderly Daycare Centres on the experiences of healthcare providers from various disciplines. Elderly Daycare Centres not only provide healthcare services that offer emotional support but also positively influence seniors' mental and physical well-being. Multidisciplinary healthcare providers caring for older adults encounter various experiences shaped by the complexities of aging populations. Support of this nature will heighten awareness of the difficulties elderly individuals encounter and advance potential solutions. Therefore,

**Funding:** Thai Health Promotion Foundation (ThaiHealth) (Grant number 65-00-0192). The funders had no role in the study design, data collection and analysis, decision to publish, or preparation of the manuscript.

**Competing interests:** The authors have declared that no completing interest exist.

healthcare professionals stressed the need for a more holistic and coordinated care framework to improve both the delivery of services and health outcomes for elderly individuals.

## 1 Introduction

The population aged 60 or older will represent 22 percent of the world's total population in 2050. Additionally, the average life expectancy for those reaching 60 years of age is going to increase to 30 years [1]. This demographic shift underscores the growing importance of elderly daycare centres and healthcare professionals in providing affordable care as essential components of sustainable healthcare systems worldwide [2]. According to the current situation, it has been estimated that the elderly population worldwide, including in Thailand, is continuously increasing. In 2040, Thailand will have an elderly population of up to 21.4 million people (31.3 percent) of the country's population [3]. The rapidly ageing population worldwide has increased pressure on health services delivery arrangements and the demand for elderly day care services.

In Thailand, the 1997 Constitution, Section 80, mandates that the state support the elderly, the impoverished, individuals with disabilities, and the underprivileged to ensure an excellent quality of life and self-reliance [4]. The 11th National Economic and Social Development Plan (2012– 2016) also emphasized the importance of community participation at all levels—local, regional, and national—in formulating strategies to achieve a society characterized by happiness, equality, fairness, and resilience to change. In terms of allowances to elderly individuals who are unable to care for themselves and lack support from their children. The state offers free medical treatment to the elderly, along with assistance and knowledge in various areas. Services include home visits by nurses, visits by volunteers and social workers to converse with older people, and establishing elderly clubs or associations. These initiatives aim to educate, clarify, and promote mental health, offering care within and outside hospital settings. The care models for older people in Thailand have been developed to address various needs, encompassing both medical and social aspects. These diverse care models aim to address the multifaceted needs of older people in Thai communities, ensuring they receive appropriate support and maintain a good quality of life. These models include home visits, Home health care, Community-based service (Day Care Centers and Elderly Clubs), Home care service, Community-based service providers, and Elderly care facilities [5].

Elderly Daycare Centres represent a crucial intervention in addressing these disparities by providing services that promote physical health, cognitive stimulation, and social connection for older adults across diverse socioeconomic backgrounds in comprehensive geriatric care [6]. Elderly care services in community day care centers primarily provide daily and spiritual comfort. The demand for these services is influenced by various factors, including age, education level, income source, availability of caregivers, satisfaction with services, and the individual's ability to perform activities of daily living (ADL). Implementing timely nursing care policies and measures that address these factors is essential to enhance elderly care [7]. The elderly adults are given care, medical assistance, emotional support, aid, advice, and a space where they meet others in similar situations. The day care centre also acts as an outlet for senior citizens to spend time and indulge in recreation activities, outings and gatherings [6].

Healthcare professionals in day care facilities for the elderly navigate a distinctive set of challenges and possibilities when providing high-quality care to older adults. These dedicated centres function as essential community assets, delivering organized therapeutic programs,

social interaction opportunities, and temporary relief for caregivers while meeting the intricate requirements of seniors who need oversight but do not require around-the-clock residential services [8,9]. The role of healthcare providers in these settings extends beyond traditional clinical responsibilities. Staff members often develop meaningful relationships with attendees, becoming attuned to subtle physical and cognitive health changes while fostering an environment that promotes dignity, independence, and quality of life. Their experiences offer valuable insights into the practical realities of geriatric care delivery and the evolving landscape of community-based elder services [9].

The rising proportion of older adults has positioned elderly daycare centers as critical components of community-based care, offering crucial services that facilitate continued independent living and mitigate the need for institutionalisation. An integrated theoretical framework drawing upon Person-Centered Care (PCC), Activity Theory (AT), and Organizational Support Theory (OST) allows for an examination of the individual, interpersonal, and organizational dimensions of care delivery. These three theories are not independent but are interdependent, thereby forming a potent and integrated framework: Person-Centered Care (PCC): McCormack's [10] initial four concepts, central to person-centered practice, have been expanded to include the crucial dimensions of being in relation, the social world, place, and self. These additions provide a more comprehensive framework for understanding and implementing person-centered care [10]. PCC is an approach to care grounded in philosophical and ethical principles. It arranges the individual by focusing on the distinctive values, preferences, needs, and inherent strengths [11]. These relationships encompass all care providers, service users, and their significant others. This approach is also anchored by core values, including respect for individuals, the right to self-determination, mutual respect, and shared understanding. PCC Framework highlights four key components: prerequisites (provider attributes), the care environment, person-centered processes, and expected outcomes. It is widely acknowledged as the standard for high-quality care in geriatric contexts. PCC also shapes the professional philosophies and goals of all care providers, including nurses, social workers, therapists, and activity coordinators. Understanding their individual perspectives is crucial, as these insights reveal how they interpret, understand, and ultimately strive to implement PCC in their daily practice. Activity Theory (AT) [12]: Activity Theory pleases as a crucial framework for comprehending successful ageing and the significant role of elderly daycare centers in fostering continued engagement among older adults. This theory emphasises that an individual's psychological and social needs persist throughout their lifespan. It further suggests that a reduction in activity during older age often stems from societal impediments rather than personal preference or a decline in ability. In the context of elderly daycare centers, healthcare providers are essential in facilitating activity for older adults. They help these individuals navigate age-related changes while simultaneously supporting their sense of purpose and social connections. Organisational Support Theory (OST) [13]: OST examines the impact of organisational factors on employee well-being, job satisfaction, and performance. It proposes that employees develop a general sense of how much their organisation values their work and well-being. In a daycare setting, the overall organizational environment—including management, available resources, and workplace culture—directly affects how well and how consistently providers can do their jobs. Ultimately, if providers feel supported by their organisation, it will boost their motivation, engagement, and willingness to excel in their roles. The WHO emphasizes that aging intersects significantly with numerous Sustainable Development Goals (SDGs), particularly in enhancing the functional abilities of older adults [14]. Specifically, Goal 3 aims to ensure healthy lives and promote well-being for all ages by achieving universal health coverage, which includes financial risk protection. Additionally, Goal 10 focuses on reducing inequalities within and among countries by fostering the social, political, and

economic inclusion of all individuals, regardless of age. The right to health is also one of the human rights, inseparably connected with the ethical principles of human dignity and social justice. This standpoint is represented and anchored in numerous international treaties, i.e., the Universal Declaration of Human Rights [15] and the Constitution of the World Health Organization [16].

Despite the rapidly increasing ageing population, healthcare services for older people in the community remain unorganised and care facilities inadequate. These challenges are limited research focusing specifically on the experiences of healthcare providers in daycare settings. Understanding the experiences of healthcare providers in the settings is crucial for improving service quality and staff retention. This study aims to enhance understanding of healthcare providers' experiences in the Elderly Daycare centres, informing improvements in practice and policy.

## 2 Materials and methods

### 2.1 Research design

Interpretative phenomenological study questioned the experiences of healthcare providers working in the Elderly Daycare Centres in order to bring out what is usually hidden human experience and in human relations. This method goes beyond the descriptive in looking for the meaning hidden in the narratives and assists in the understanding of how individuals make sense of their work experience as well as understanding the perspectives of multidisciplinary healthcare providers in Elderly Daycare Centres.

Using purposive sampling, 13 healthcare providers (e.g. nurses, healthcare professionals and public health technical officers) delivering care to older people at the Elderly Daycare Centres were interviewed in Thai. They was also done based on their experiences working in Elderly Daycare Center. Informed consent was taken from the participants before the interviews. Face- to- face interviews were held with all participants in a private room at their workplace in order to support their comfort in sharing personal experiences and insights about their work. Inclusion criteria: o To be the core multidisciplinary team (nurses, care assistants, allied health professionals) o Have been employed in Chiangrai Municipality o Responsible for maintaining consistent care services in the Elderly Day Care Centres. Exclusion criteria: o External consultants to the Elderly Day Care Centres.

### 2.2 Setting

All healthcare providers from four Elderly Daycare Centres in Chiangrai municipality, Chiangrai province, Thailand were selected to share their experiences in delivering care to older people.

### 2.3 Data collection

This study protocol was approved by the University of Phayao Human Ethics Committee (Thailand) prior to the commencement of data collection (HREC- UP- HSST 1.2/043/67). Prior to conducting interviews, the researchers approached the management of elderly daycare service to understand management perspectives in elderly care settings. The principle researcher (PO) contacted potential participants to ask the screening questions again to verify that potential participants met the inclusion criteria and to confirm their willingness to take part. Participants gave their informed consent to participate voluntarily after receiving full explanation about their right to anonymity and to refuse or subsequently withdraw from the study. Written informed consent was obtained from all participants using standardised

consent forms which were signed and dated by both the participant and the principle investigator (PO) prior to the arranging and conducting interviews. In-depth semi-structured interviews were conducted using an interview guide.

The development of our interview guide followed a structured, multi-stage approach that drew upon established theoretical foundations while remaining attentive to real-world applications. We began by undertaking an extensive review of existing literature on collaborative practice among healthcare professionals working with older adults. This review revealed five critical areas that shaped our initial question development: how professionals understand and articulate their roles, the establishment of frameworks for overseeing healthcare delivery systems, the capacity and effectiveness of organisations, challenges related to distributing resources, and outcomes to furnish to the older persons.

Open-ended narratives also captured the depth of individual experiences. The interviews were recorded using a recorder. The interviews lasted around 60 to 90 minutes each. Filed notes were written after observation in each interview. The interviews were conducted between 24 March 2024 to 30 June 2024. No further interviews were conducted as soon as data saturation was reached.

## 2.4 Reflexivity

This reflexive examination of our positionality as Thai geriatric nurse researchers reveals how professional expertise, cultural background, and linguistic context simultaneously strengthened and potentially biased our research approach. Our nursing specialisation provided insider understanding of multidisciplinary healthcare dynamics while potentially creating bias toward nursing perspectives and collaborative care assumptions. Prior to undertaking this study, the researchers developed preliminary conceptual models of elder daycare management, informed by a review of existing literature in healthcare management, social care and ageing theory. The literature consistently emphasises that effective management in geriatric care settings plays a critical role in determining the quality of care and services provided to older adults. Thai cultural values emphasizing elder respect, family-centred care, and hierarchical relationships may have predisposed us toward viewing certain care approaches favourably. The researchers not only conducted this research in Thai language facilitated authentic participant expression but also created shared cultural assumptions requiring careful examination. Our policy background enhanced understanding of systemic influences while potentially over-emphasising structural factors in interpretation.

To address these biases, we implemented systematic mitigation strategies including reflexive journaling, interdisciplinary peer debriefing, discipline-specific member checking, disconfirming evidence searches, and cultural reflexivity practices. These approaches enhanced research trustworthiness while maintaining transparency about our interpretive influence. Our cultural and professional insider status, when combined with systematic bias awareness and mitigation, enhanced rather than compromised research quality by enabling nuanced understanding while maintaining analytical rigour. This transparency provides readers with clear context for interpreting our findings within the Thai healthcare environment while contributing valuable insights to international geriatric care literature.

## 2.5 Data analysis

The ATLAS.ti software program was used to manage the data and coding. The interviews were conducted in Thai and transcribed verbatim. Employing the hermeneutic circle as a methodological cornerstone, a foundational principle of Interpretative Phenomenological Analysis (IPA), the researchers engaged in an ongoing interpretive process that involved a reciprocal

movement individual accounts. This process allowed for a deeper understanding of how multidisciplinary healthcare providers interpret their experiences in the Elderly Daycare centres, particularly in relation to team dynamics, professional roles, and institutional structures. The data was analysed using Interpretative Phenomenological Analysis (IPA) followed the steps initially identified by Manen [17] and Smith, Flowers & Larkin [18]:

The initial phase of the analysis involved a continuous process of reading and rereading the transcripts, which was sustained throughout the entire data analysis. During this process, the principle researcher consistently revisited the original texts while engaging in coding and interpretation, critically reflecting on the data by asking questions such as, "What does this reveal about the phenomenon?" and "How is this connected to geriatric care?" This approach aligns with interpretive phenomenological methodology to enhance the depth and rigour of the analysis and to ensure a clear audit trail. The principle researcher identified meaning units within each interview transcript. These texts were meticulously examined to capture the participants' perspectives on the phenomenon under investigation. Individual attitudes and experiences were precisely coded, and codes reflecting related aspects of the phenomenon were then grouped into relating sub-themes. During the coding process, a funnelling approach was applied to manage the large volume of initial codes, allowing for the refinement and exclusion of codes deemed unrelated or lacking analytical relevance at later stages. Analytical memos were used to document reflections drawn from field notes and the reflexive journal for each interview, capturing insights from the day of data collection to develop a comprehensive understanding of each healthcare providers' experience.

Through further analysis, the codes appeared to be saying the same thing about particular phenomena. These codes were grouped under certain categories or sub-themes based on the importance of aim and objectives. Once these sub-themes were explored, facilitating the emergence of broader themes that provided a coherent and nuanced interpretation of participants' experiences. A member-checking process was employed to clarify any discrepancies in interpretation, during which feedback and suggestions were solicited from the interpretation team on a finalised draft.

## 2.6 The rigour of the study

Data reliability and validity were assessed using the concepts of credibility, dependability, confirmability and transferability in increasing the trustworthiness of this study.

Credibility: Purposive sampling was used to choose individuals who possessed direct experiences of the phenomena under investigation and were eager to share their narratives, ensuring credibility. Building trust can also be achieved through the participant and researcher developing a relationship throughout the course of the interview. To ensure accuracy and credibility of the data, member checking was conducted by returning interview transcripts to participants, who were invited to review and correct any mistakes or misinterpretations.The hermeneutic process presented challenges, particularly in determining when interpretation had reached sufficient depth. The concept of 'interpretive saturation'—defined as the point at which successive analyses yield no new or meaningful insights. This was operationalized through repeated interpretive cycles, and saturation was considered achieved.

Dependability: An external evaluator to assess the analysed data, ensuring that no critical elements were overlooked during the data collection and analysis process. Data from several data collection processes such as interviews, observations and written field notes were used and all analytical data were linked to serve understanding of the phenomena of geriatric care in Elderly Daycare Centres.

Confirmability: The principle researcher maintained a reflexive journal, recording thoughts and reflections both before and immediately after each interview, as well as taking notes during the interviews. This practice aimed to mitigate potential biases. The reflexive journal served as a tool to document the researcher's evolving insights, emotions, and conceptualisations throughout the study, contributing to the transparency and rigour of the research process. This level of transparency enables readers to form their own interpretations of the findings and apply their own preconceptions to the analysis. Furthermore, two experts in qualitative research independently audited all aspects of the research process, documentation, and preliminary findings to ensure the credibility and dependability of the study.

Transferability: While the findings of this study are context-specific and not inherently generalisable, they are presented in a manner that enables readers to assess their relevance and potentially apply them meaningfully to their own experiences and professional contexts. During the research process, the principal researcher recognised that certain experiences described by participants resembled those previously observed in other healthcare providers prior to the study. This resemblance was deemed valid and suggests that the insights shared by participants may hold relevance for other multidisciplinary healthcare professionals working with similar populations.

## 3 Results

### 3.1 Demographic characteristics

A total of 13 healthcare providers from four Elderly Daycare Centres participated in this study in Chiangrai Municipality, Chiangrai province. Most participants were female (84.62 percent) and male (15.38 percent), with the ages of 20 to 30 (38.46 percent), 31 to 40 (38.46 percent), and 41 to 50 (23.08 percent). Participants represented a broad range of primary healthcare providers, including Registered Nurses (61.54 percent), Healthcare professionals (23.08 percent) and public health technical officers (15.38 percent). Most participants had less than 4 years of experience (76.92 percent) and over than 10 years of experience (23.08 percent) in their current working role (Table 1).

**Table 1. Demographic details of participants.**

| Participants | N | % |
|---|---|---|
| **Position** | | |
| - Registered Nurse | 8 | 61.54 |
| - Healthcare professional | 3 | 23.08 |
| - Public health technical officer | 2 | 15.38 |
| **Gender** | | |
| - Female | 11 | 84.62 |
| - Male | 2 | 15.38 |
| **Age (year)** | | |
| - 20-30 | 5 | 38.46 |
| - 31-40 | 5 | 38.46 |
| - 41-50 | 3 | 23.08 |
| **Experience** | | |
| - less than 4 years | 10 | 76.92 |
| - over than 10 years | 3 | 23.08 |

## 3.2 Perspectives of multidisciplinary healthcare providers in Elderly Daycare Centres

The initial two programs were created in four Elderly Daycare centres, such as the muscle strengthening program and the nutrition program for individuals with nutritional problems.

**3.2.1 Theme 1: Opportunities in policy implementation.**   Healthcare providers viewed government policies as flexible guides for their care, not as rigid rules. This finding shows that structured reflection is a key part of Organizational Support Theory. Instead of following mandates as unchangeable commands, providers used them as a starting point, modifying them based on older adults' needs, available resources, and their own judgment. This approach made providers feel more valued and supported, boosting morale and their sense of purpose in their work. It perfectly illustrated how the mandate was not a directive to be followed blindly, but a guide to be applied thoughtfully, with the ultimate goal of older adults well-being in mind.

**Health service delivery.**   The initial point of providing care for older adults in Elderly Daycare Centres is to implement a policy to provide services for older people and allow them access to comprehensive care services.

*The policy has been implemented into a program with clear weekly work assignments. We hold meetings before activities begin and weekly review meetings to discuss the outcomes of activities, which helps us identify strengths and areas for improvement, allowing us to adjust for better results* (Nurse 2)

*The management's policy aimed at reducing inequality among older people has been transformed into a practical program with visible physical results and impacts on the mental well-being of elderly participants who engage in social activities* (Nurse 3)

**Providing multidisciplinary care services.**   The experiences of healthcare providers at the Elderly Daycare Centres, as shared through a series of in-depth interviews, revealed a complex and nuanced relationship with the government mandates that govern their daily practice. The providers didn't ignoring government mandates, but instead are using their professional judgment to adapt them so they can provide the best possible care for each person. They explained how challenging it is to apply a single set of rules to a diverse group of older adults, each with unique needs. The providers felt a deep sense of responsibility to find a balance between keeping everyone safe and respecting the individual dignity and needs of the people they care for. This connects directly to the core tenets of OST, which is about workers' generalized belief that the organization values their contributions and cares about their well-being.

Healthcare providers in primary care defined how they used a collaborative team approach to provide holistic care for older adults through programs. All providers included various specialists, such as nurses, healthcare professionals and public health technical officers, who worked together to ensure holistic treatment and support for older adults through coordinated programs.

*I was assigned to organize movement programs and design exercises to increase muscle strength in older people and improve their balance. The recreational activities are collaboratively managed by nurses and public health academics. Other staff members at the centre also help with the program whenever they are available* (Healthcare professional 1)

*There is joint planning for activities, with group meetings held before and after each session to review results and improve future sessions. In the pre-event meeting, we set up the venue, coordinated with relevant parties, and confirmed attendance with the elderly participants. The activities involve collaboration with various healthcare professionals, such as public health academics,*

*nutritionists, physiotherapists, and other staff at the center who assist with registration, as well as snack and lunch preparation.* (Nurse 2)

**Continuous proactive healthcare service.**  Providing healthcare services in two programs implemented preventive measures and health promotion activities. The group health activities within these programs created meaningful experiences for participants, fostering social connections while enabling them to gain knowledge through interactive learning sessions.

To ensure care delivery can flex and adapt, well-organised weekly team meetings were comprehensive Clinical Learning Systems.During these multidisciplinary gatherings, healthcare providers collectively reviewed the previous week's care outcomes. This process involved a thoughtful reflection on actions taken and their subsequent impact, fostering both professional growth and a deeper sense of purpose in their practice. The implementation of regular weekly planning and reflection sessions created a continuous feedback loop that substantially enhanced the quality and consistency of care protocol delivery.

*We develop strategies each week to prevent the elderly from getting bored or losing interest in the program by organizing fun recreational activities that are different each week. We use the LINE application to communicate with older people and provide videos for them to review their exercise routines at home. If elderly participants cannot attend the program, they can review what they have already learned to build upon their training in the following week.* (Healthcare professional 1)

*The strategy used in organising recreational activities and providing knowledge that the elderly need has proven practical and effective. This makes the program interesting, and older people remain interested and enthusiastic about participating in activities continuously.* (Nurse 1)

**Created community healthcare networks.**  Community healthcare volunteers and caregivers (CGs) work as the community health network. They demonstrated adequate capacity and capabilities to effectively support and sustain the program's activities. They also aided during exercise and nutrition sessions.

*Community healthcare volunteers have varying levels of capability, while others are not. Participation in the program is voluntary, but most are cooperative in organizing activities. Overall, the community health network has sufficient capacity to support the program's activities.* (Nurse 1)

*Community healthcare volunteers and caregivers (CGs) assist in movement programs by supporting older people during exercises, ensuring the safety of those at risk of falling, and monitoring elderly participants. They also help follow up with elderly individuals who cannot be reached.* (Healthcare professional 1)

**3.2.2 Theme 2: Challenging experience.**  Delivered healthcare service in two programs was a novel experience for healthcare providers, particularly in-home care visits.

*As a new nurse, I was given the opportunity to work as a program manager, presenting the program to colleagues and listening to everyone's opinions. It was a challenging experience because I had never done it before. There were both fun moments and problems to solve, and it was something new to learn from.* (Nurse 1)

*During each week's program activities, we provide education about movement, exercise, and medication knowledge, along with recreational activities to keep the atmosphere from becoming too tense. The three of us also conduct home visits to assess environmental risks within their homes that could lead to falls and accidents. Our home visits were new experiences of my duties. I felt I could do my duties outside the workplace.* (Healthcare professional 1)

**Transforming heavy work into entertaining activities.**  *As healthcare providers carried out their duties in a way that made them feel less burdensome, these responsibilities transformed into rewarding aspects of their work and became more like enjoyable performances. After being*

*assigned to implement program activities, I've grown to enjoy and have fun with the work. I've become someone who enjoys doing activities, even though I didn't like them before.* (Public Health Academic Officer 1)

*The elderly said that it's not difficult at all to choose both the type and appropriate amount of food for themselves. For us as service providers, what started as feeling tired and worried about lack of cooperation has transformed into having fun every time we organize activities. The positive outcomes we see in the elderly have become our motivation. The elderly are encouraged to continue and are willing to participate in every activities.* (Healthcare professional 2)

**3.2.3 Theme 3: Enhancing organisational effectiveness and engaging support.** The findings from this study reveal that the organisational framework of the Elderly Daycare Centres significantly impacts staff motivation and the effectiveness of interprofessional collaboration. Elderly Daycare Centres managed by a director, with distinct teams for nursing, caregiving, and social work. Within this structure, nurses and other healthcare providers tend to operate with a high degree of autonomy, independently managing their respective professional duties. Nurses and other healthcare providers operated largely independently within their own professional roles and took responsibility for different aspects of this holistic care plan.

In essence, interpretation reveals that institutional structures are not neutral containers for healthcare delivery but active forces that shape the professional experiences, motivations, and collaborative capacities of healthcare providers. Effective institutional design, therefore, requires a recognition of this dynamic relationship and the intentional creation of structures that support both individual professional fulfillment and collective effectiveness in older adults' care.

Improving an organisation's ability to operate effectively and achieve programs' goals involves better resource management, staff training, technology adoption, and streamlined operations to maximize efficiency and effectiveness. The programs combine education, recreation, and hands-on practice to create an enjoyable and beneficial experience for elderly participants. Family involvement further encourages participation, fostering social engagement and improved health. Regular follow-ups, home visits, and pharmacist support enhance the program's effectiveness, ensuring that elderly individuals receive continuous care and guidance.

*The program integrates recreational activities with education and hands-on practice, making the sessions enjoyable.The elderly particularly enjoy recreational activities. There are review sessions and supplementary activities for elderly participants who miss some weeks, ensuring they receive the program's full benefits. However, only a few miss sessions, and they usually return to the program as usual. Families also play a role in encouraging older people to join the activities, as they want them to engage socially, avoid loneliness, and improve their health.* (Nurse 3)

*The duration of each program is appropriate, with regular follow-ups and scheduling through the Line application. During the program, home visits are conducted for periodic evaluations. I also visited to monitor medication use, helping elderly participants better understand their medications.* (Healthcare professional 3)

**Knowledge management.** *Nurses and academic staff are primarily responsible for leading activities. Some community health volunteers or caregivers (CGs) can conduct recreational activities, but only a few. Some are hesitant or have never done it before, making it more challenging.* (Nurse 7)

*The team has sufficient knowledge, including nurses, public health academics, and a pharmacist who provides excellent medication guidance. Before each program, we meet to discuss and*

*assign tasks, determining who will be the main speaker and who will lead the recreational activities that week. By working together, no single person is overwhelmed. Over time, we found it enjoyable, and the elderly participants had fun with us, too.* (Healthcare professional 1)

**Program-based training for elderly care.** Encouragement of training and regulatory practices for healthcare providers fosters continuous self- improvement, enabling them to deliver enhanced care services for the elderly. The participants emphasised the importance of training a diverse group of health and general personnel to manage the care of older individuals, particularly in response to health challenges. Healthcare providers should prioritise addressing the health concerns of older adults by developing the necessary skills and competencies to offer effective care.

*Before implementing the program, we collaborated with educational institutions to train on the use of the program, as well as to assess and screen elderly individuals for participation. This increased knowledge and boosted confidence in delivering healthcare services according to the specific programs outlined. Additionally, home visits were conducted by a multidisciplinary team, including nurses, public health professionals, nutritionist, physical therapist, and pharmacist. This collaboration led to a clear and unified goal in promoting the health of elderly individuals at risk of falls and those with nutritional issues.* (Nurse 8)

*I participated in the program, which was guided by experts from educational institutions to assess elderly individuals at risk of falls. The program included activities to strengthen muscles, followed by the implementation of the planned activities and post-program evaluations. The training made the role of the physical therapist clearer and more defined, and it instilled a sense of pride in fulfilling my professional role in caring for elderly individuals at risk.* (Healthcare professional 1)

**Facility accessibility and adaptations.** This category focuses on the challenges and adjustments made to accommodate elderly program participants. Although the space is limited and not fully accessible, modifications are being made to enhance the facility's usability. These include adjusting the layout for specific activities and ongoing renovations to improve the activity space, such as converting it into a glass-enclosed room.

*The center has made improvements to the facility to accommodate the program better. However, space is somewhat limited. The room has air conditioning and fans, but no handrails or ramps make it difficult for elderly participants to enter. The restrooms are in separate buildings not too far, but for elderly individuals with mobility issues, community health volunteers need to assist them.* (Nurse 7)

*Initially, the program was conducted in an open space, but when the weather became too hot, we moved to an air-conditioned room on the second floor. There were concerns about whether the elderly participants could access it easily. However, they were willing and happy to go upstairs. Since then, we have continued holding activities in the second-floor room.* (Healthcare professional 1)

**3.2.4 Theme 4: Reflections on program outcomes and impacts.** Enhancing accessibility and participation for the Elderly Providing transportation services for elderly individuals enhances their ability to participate in activities, reducing barriers related to mobility, cost, and dependence on others. Consistent participation in these programs leads to noticeable improvements in physical health and mobility, encouraging more elderly individuals to join and benefit from the experience.

*The transportation service for elderly individuals allows those who cannot travel on their own to participate in activities consistently. Without this service, many elderly individuals would miss the opportunity to join the program.* (Nurse 1)

*Having a transportation service for older people makes it easier for them to join activities without worrying about travel, costs, or relying on family members for transportation.* (Nurse 3)

**Promoting physical and mental well-being.** Healthcare providers described the improvements in the physical health and mental well-being of elderly participants. The program also led to increased mobility, weight changes, and better blood sugar levels, resulting in a sense of satisfaction and encouragement to others to join. Older people also experienced mental benefits, feeling motivated by their progress. Healthcare providers also benefited by expanding their knowledge and skills. Additionally, strong collaboration from other centre staffs, who assisted with registration, venue setup, and maintaining order, contributed to the program's success.

*The outcomes can be seen in two aspects. First, the elderly participants experienced significant physical and mental improvements; they felt impressed and encouraged others to join the program. Second, the service providers also benefited by expanding and developing their own knowledge. Additionally, there was strong cooperation from other centre staffs who helped with registration, venue setup, and maintaining order.* (Nurse 2)

*The elderly participants who joined the program showed clear improvements in their daily lives. For example, those who previously had difficulty walking could walk more easily and even drive after participating in the program. Some who experience positive results from participating in the program encourage others to join because they notice improvements in their physical condition and mobility after joining.* (Nurse 6)

Maintaining a sharp mental faculty is as vital as being physically active.

*The outcomes reflected by the elderly participants show the benefits they gained from the program. Their happiness comes from improved mobility. Those who once struggled to walk can now move more easily. They also enjoy meeting others with similar challenges, conversing, and receiving valuable advice. As for the service providers, our happiness comes from seeing older people change their behaviors, improve their mobility, lose weight, and lower their blood sugar levels.* (Nurse 3)

*The program activities have motivated elderly individuals who were initially reluctant to join. Through participation, they become more engaged, meet other elderly individuals, and have the opportunity to see physiotherapist and pharmacist. They don't just seek knowledge or exercise; they also want to learn about other topics and leave with both knowledge and peace of mind. As service providers, we also feel fulfilled and happy seeing this impact.* (Healthcare professional 1)

## 4 Discussion

The study aimed to understand delivering geriatric care of healthcare providers working in Elderly Daycare Centres to gain insight into their perspective of service caring for older people. Study findings provide a crucial new understanding of the experiences of this group of healthcare providers (e.g., registered nurses, a physiotherapist, a nutritionist, public health technical officers, and others) in healthcare services for these individuals. According to healthcare providers, the important factors to consider in the care of older people are preventive programs, the training of multidisciplinary healthcare personnel on healthcare problems specific to older persons, and facilities for older persons.

### 4.1 Increasing access to healthcare services and reducing inequality for the elderly

The findings of this study underscore the critical role of Person-Centered Care (PCC) in shaping the professional attitudes and practices of healthcare providers within elderly daycare centers. The data revealed how providers' understanding and interpretation of PCC directly translate into the successful implementation of active aging programs, ultimately enhancing the well-being and autonomy of older adults (Fig 1).

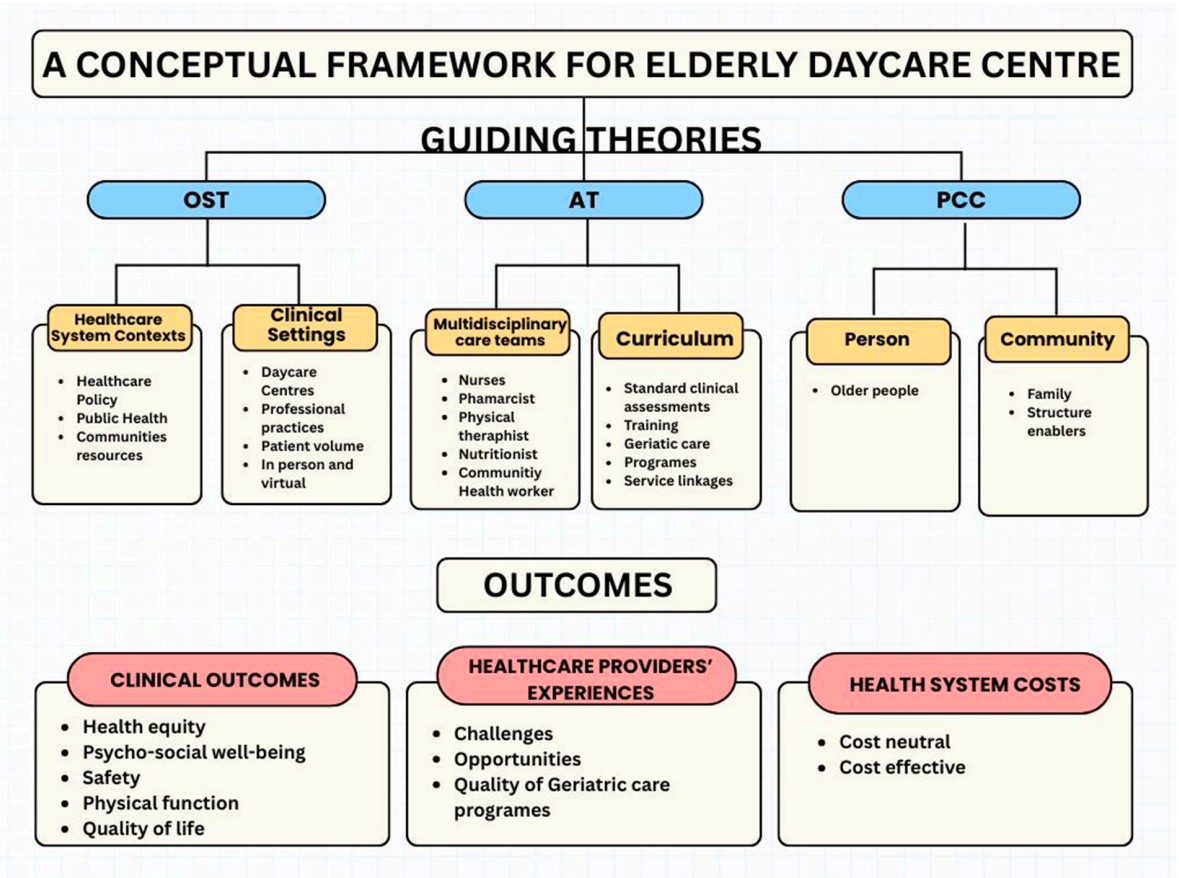

**Fig 1. A conceptual framework for the Elderly Daycare Centre.**

A core finding of this research was that successful active aging programs were founded on a person-centered approach that valued the individual above all else. Healthcare providers conducted individualised assessments to design activities that were not only meaningful and engaging but also honored each person's unique history and abilities. This approach, centered on respect and dignity, empowered older adults to actively participate in choices that enhanced their satisfaction and quality of life [19]. For instance, the collaboration among multidisciplinary teams ensured that care was tailored to individual needs; nurses kept activity coordinators informed of health changes, while a pharmacist took the time to explain medications, empowering older adults to make informed decisions about their own non-urgent care.

The implementation of PCC in these Daycare Centres was instrumental in giving older adults more control over their own lives. The goal was not merely to meet immediate health needs, but to empower them to make choices and feel a sense of purpose. Similarly, empowering older adults help to challenge ageism by highlighting the strengths, capabilities and future opportunities as well as facilitating patient-centered care [20]. It was made possible by a collaborative environment where healthcare providers recognized and built upon each person's unique strengths. This interdisciplinary effort demonstrated how the ethical and philosophical tenets of PCC can be operationalised to create an environment where older adults are not passive recipients of care, but active participants in their own well-being [21].

Elderly Daycare Centres also enable older people at risk of falls and who had nutrition problems to access health services in terms of support facilities, program activities, and home visits from multidisciplinary healthcare professionals. The service programs meet the needs of older people and their relatives. To enhance the well-being of an aging population, it's crucial to both make healthcare accessibility and the utilisation of healthcare services [2,22]. Healthcare providers employed strategies to reduce costs related to medications, transportation, and support services that would assist older adults in better managing the complexity associated with the risk of falls and nutrition. In other words, this supported raising access to healthcare services. This aligns with previous studies that supported the idea that primary care professionals play a significant role as medication optimisation specialists, taking responsibility for adjusting treatments, implementing de-prescribing when appropriate, and creating personalised approaches to enhance medication benefits [22,23].

Previous studies found that healthcare accessibility was compromised by multiple transportation-related factors: no private vehicle access, undependable public transportation systems, extended travel distances to providers, financial burdens of travel, and physical limitations affecting mobility [23]. Not only transportation related costs to healthcare appointments obstacles, but the inability to operate transportation options due to disability was also a transportation barrier to accessing healthcare services [23]. However, in this study, facility service in terms of transportation can support healthcare access. Similarly, healthcare accessibility for older adults was being addressed through targeted interventions: strengthening workforce capabilities, optimising facility resources, and enhancing information management practices [24]. Furthermore, enhancing access to necessary public health services for the older population was a significant challenge and an obstacle on the road [24]. Supporting older adults in overcoming disability-related barriers to healthcare access serves as a means of social inclusion, allowing them to participate physically in muscle-strengthening programs. This was one of the solutions presented in studies for dealing with the physical limitations, which was to encourage movement and to exercise safely and efficiently [22,24].

Eliminating health disparities and promoting health equity are to address interconnected public health challenges [25]. Although not all health disparities would be eliminated, policies and practices that lessen or eradicate the underlying causes of health inequities and healthcare disparities would significantly reduce them and bring the country closer to health parity. The current study is unique as it provided improving access to healthcare services in Elderly Daycare Centres, one of the strategies to encourage equal opportunities in healthcare. Considering the healthcare providers in the study, they felt that accomplishments in healthcare services could assist older adults in accessing healthcare services and reduce health disparities, boosting the older population's overall well-being.

Interestingly, the finding was consistent with previous studies which pointed out that healthcare providers are responsible for addressing inequalities and working towards a more equitable healthcare service [25,26]. Achieving health equity includes the principle of fairness in health systems for all individuals, especially elderly adults [25,27]. For the Convention of Human Rights for Older People, the UN has recently brought attention to the unique problems facing the elderly and the need for policies that improve access to care and support, including long-term care, and fortify the healthcare and social protection systems [27]. The findings suggested that successful elderly care is not just about the programs offered, but about the deeply rooted professional attitudes and collaborative efforts of providers who view their role as empowering individuals to be active participants in their own lives.

## 4.2 Improving the quality of life of older adults

Quality of life plays a key moderating role between physical activity and essential factors like psychological resilience, mental health, and health behaviours [28,29]. These multiple dimensions can improve the Quality of Life and well-being [20]. Collaboration across multiple disciplines plays a crucial role in improving the quality of life for older primary care patients [8]. In this study, healthcare professionals often encouraged elderly clients to engage in structured social and cognitive activities, aligning with Activity Theory, which posits that sustained engagement contributes to improved well-being in old age. They reported that facilitating these interactions provided observable benefits to client mood and cognition, reinforcing their satisfaction.

Elderly Daycare Centres offered recovery programmes, social interaction with peers, and participation in programs, meals, and physical activities. They also supported physical and mental well-being health including the provision of purposeful activities. These can reduce social isolation among elders and provided enjoyment and a good environment in their family and home [6]. Healthcare professionals described that they encouraged older adults with the programs in terms of strengthening muscle program and nutritional health program. They could better manage their own healthcare. Such activities in Elderly Daycare centres were expected to improve the quality of life of older people. This study also found similar results that Daycare Centres in the therapies offered preserve improve the quality of life, social life, physical function, and mental health of older attendees. They also provide social interaction, activities, and therapies that enhance life quality, assist in managing pre-existing diseases, and may stop deteriorating health and function [28].

Getting out of home was also a chance to socialise and enjoyment with a community-based group [6]. Social participation promoted well-being and reduced isolation and loneliness [30]. Healthcare providers in the study offered supporting ageing in place through well-being and education. Therefore, seniors' ability to health life's difficulties affects their overall, demonstrating how quality of life links physical activity to improves psychological wellness. Previous studies on primary care providers had recognised the engagement in physical activity yields significant positive outcomes across multiple dimensions of health and well-being, mental well-being, and psychological resilience [29–31]. Encouraging physical and psychological well-being as well as social activities can also improve their quality of life and health.

The association between physical health, psychological well-being, and social connection also reveals the quality-of-life benefits of older adults in Daycare Centres [32,33]. Daycare service centres that incorporate structured physical activity protocols demonstrate statistically significant, albeit moderate, enhancements in participants' functional capabilities, postural stability, and capacity to perform essential self-care tasks. These improvements correlate with quantifiable reductions in frailty indices and fall probability metrics, suggesting a protective effect against age-related functional decline [33]. This substantial reduction in psychological distress indicators suggested that structured engagement in community-based day programs may function as an effective non-pharmacological intervention for geriatric mental health management [34]. Subsequently, Orellana [35] revealed statistically significant enhancements in participants' perception of available social resources and interpersonal relationship quality. The findings suggest that the structured social engagement opportunities inherent in day centre environments contribute substantially to rebuilding and maintaining essential social networks that frequently diminish during advanced ageing. Similarly, Albarqi [8] recommended that healthcare systems should focus on establishing collaborative care models, encouraging interdisciplinary cooperation and implementing strategies that enhance social connections for the ageing population.

## 4.3 Challenging and rewards in healthcare services for older adults

The results showed that the provision of knowledge support and training for assessing the elderly was a significant factor. This finding directly supported OST, which suggested that an organisation's resources can either enable or constrain healthcare providers' ability to implement specific approaches. Likewise, the training provided empowering the healthcare providers to apply their knowledge and skills, thereby facilitating the implementation of activity-based and person-centered care [36,37].

Furthermore, the support for developing specialised programs, such as exercise programs for improving muscle strength, highlights how organizational backing promotes providers' expertise and autonomy. This was consistent with OST's view that a supportive environment encouraged providers to take ownership of their roles, leading to increased job satisfaction and a more collaborative work environment [13]. This sense of being supported by the organisation—getting the tools and training they needed—directly resulted in a number of positive outcomes. Healthcare providers were more satisfied with their works and better equipped to handle the daily stresses of working in elderly daycare centers. This confirmed that when healthcare providers felt supported, they became more committed and were better able to handle the challenges of their work, an opinion that OST had consistently expressed (Fig 1).

Essentially, this study provided empirical evidence for the application of Organizational Support Theory (OST) within the specific context of elderly daycare centers. The findings underscored the crucial role of organisational support—in the form of training, resources, and autonomy—in achieving positive geriatric care outcomes. By empowering healthcare providers with the necessary tools and knowledge, organisations can not only improve providers' work satisfaction and commitment but also enhanced the quality of person-centered care for the elderly [38].

Complexly raising the growth of elderly patients and the requirements of caring frequently provide challenges for healthcare personnel [39]. Healthcare providers in this study described feelings of emotional challenging and rewards due to the demanding nature of providing continuous cognitive stimulation, personal care, and emotional support to older people. Simultaneously, many derived a deep sense of purpose and connection from their roles. Multi dimensions of improvements of Quality of Life and well-being fulfil a critical respite function that generates demonstrable positive outcomes for family caregivers, creating a complementary effect where both care recipients and their informal support networks experience meaningful advantages from participation [28]. This bidirectional benefit pattern suggested that day services effectively address the interconnected needs of older adults and those who provided daily care, representing a particularly efficient intervention within the continuum of community-based elder services. [34]. Inadequate competency and lack of integration and coordination of services were concerned in home care services for older adults [40]. Mobasseri et al. [40] recommended that improvements in healthcare for older persons were fostering partnerships among key stakeholders to strengthen healthcare delivery for older people. Providing additional training to long-term health providers or staff, raising the health awareness of older people, and teaching life skills were several important aspects of caring for older people [41]. Effective collaboration among healthcare providers and older adults' interactions created favourable conditions for meaningful. It was essential to note that healthcare providers' reflections on experiences providing care following programs encompassed encouraging recovery and rehabilitation for older people who participated in all programs.

Healthcare providers consistently highlighted the value of having supportive leadership and well-defined role expectations. Drawing on Organisational Support Theory (OST), the

perception of managerial support played a crucial role in fostering their resilience and dedication to providing care [42]. Feelings of being appreciated and included in decision-making processes were closely linked to increased motivation and a greater likelihood of staff performance [43]. In the context of Elderly Daycare Centres, OST can profoundly influence healthcare providers' job satisfaction, commitment, and their capacity to cope with the demands of their roles. The goal of professional training programs was to combine theory and practice. These included improving abilities in role fulfilment, personal-centered relationships, and elder care. Prior research indicated that trainees' self-thinking and adaptability during the training process should be enhanced, and the practicality of theoretical knowledge should be highlighted [9,40]. Enhancing interdisciplinary healthcare practitioners' professional expertise and abilities will also increase their incentive to engage in the chances, challenges, and initiatives [39,40]. To improve the professional skills of interdisciplinary healthcare practitioners, it would also be beneficial to offer evidence-based training by fusing literature with experience.

Activity Theory illuminates how multidisciplinary healthcare providers facilitate meaningful engagement through diverse programming, social interaction opportunities, and therapeutic interventions [12] (Fig 1). The theory suggested that healthcare providers play a crucial role as facilitators of activity, helping older adults navigate age-related changes while maintaining their sense of purpose and social connection. In this sense, this study uniquely uncovered healthcare providers who were experienced in healthcare services for older adults. They identified two main rewards: realising improvement in older adults' health, functional abilities, and quality of life. Previous literature in focused on caring for older people supported and reported finding satisfaction in improvements in the health and quality of life of their elderly participants and feeling valued through the appreciation expressed by these older adults [9]. The rewards in caring for older people with multiple chronic conditions in a study by Ploeg et al. [23] were a strong sense of duty or responsibility toward addressing their participants' needs. However, their study pointed out that they frequently faced significant burdens or strains due to providing this care.

The findings in the study were also consistent with challenges reported in other literature, which extended the rewards by experiences across a wider variety of community-based providers who care for older adults with multiple conditions through developing and improving their professional skills and knowledge and building meaningful relationships with both their patients and fellow healthcare providers. Healthcare providers with extensive experience in geriatric services participated in this study and had the same experiences with others. This presented taking on the challenge of complex care needs when working with older adults who have multiple chronic conditions [23].

Complex caring for patients need access to ongoing inter-professional education and training in various modalities, such as inter-professional lectures and workplace training sessions that integrated several healthcare disciplines [39,43]. The study also combined with the perspective of multidisciplinary healthcare teams about the value of implementing successful both programs for older people in Elderly Daycare Centres. In other words, that was the understanding of challenges and rewards experiences by healthcare providers for older adults and strengthening multidisciplinary healthcare team works. Furthermore, a comprehensive approach combining individual and organizational strategies was essential, recognizing that effective collaborative practice environments emerged from the joint contributions of healthcare professionals, care organizations, and health system structures working together [44].

## 5 Conclusion

Healthcare professionals working in the Elderly Daycare Centres navigated complex emotional, cognitive, and organizational challenges daily. Understanding their lived experiences offered valuable insight into how care-giving dynamics intersect with theoretical models of ageing, coping, and organizational behaviour.

Elderly Daycare Centres were designed to support older adults who experienced various challenges, such as individuals at risk of falls and those with nutritional issues. They also need essential nursing services or health monitoring during the day. These centres provided a structured environment where seniors can engage in social and recreational activities, receive necessary health services, and maintain a sense of independence, thereby enhancing their overall quality of life. Healthcare providers were employing various effective strategies to assist older adults in managing their health. These approaches included implementing person-centered care, collaborating with other providers, and addressing social determinants of health. The present study's findings suggested that the Elderly Daycare Centres were essential for creating a sense of context and belongingness amongst older people. Healthcare providers identified the necessity for a more comprehensive and integrated care system to enhance service delivery and outcomes for older people.

Healthcare providers in this study operated at the critical intersection where population needs, system capacities, and policy expectations converge in daily practice, experiencing policy requirements not as abstract mandates but as practice realities that directly shaped their clinical decision-making and inter-professional relationships. The findings revealed that healthcare providers did not experience these elements as separate contextual factors but rather as an integrated reality requiring holistic responses that bridged the gap between policy intention and clinical implementation.

## 6 Implications for practice and policy

Understanding healthcare professionals' experiences through the lens of ageing and organizational theory highlights the need for training programs that address both client-centered care and caregiver well-being. Future research should explore how institutional policies can better support adaptive coping strategies in high-demand care environments and could examine the differences between providers by profession, and by geographic setting. Selection bias may have been introduced through recruitment through authors' personal networks.

## 7 Limitation

The cultural context of the healthcare settings examined may not reflect the diversity of cultural approaches to healthcare delivery found in other regions or populations.

Regional factors, including healthcare infrastructure, resource availability, and local health policies, create specific operational environments that may limit the generalizability of findings to different geographic contexts.

The organisational characteristics of the institutions involved—including their size, structure, and institutional culture—represent particular healthcare delivery models that may not be representative of the broader healthcare landscape. These contextual factors collectively suggest that provider experiences are deeply embedded within their specific practice environments, and findings should be interpreted with consideration of these cultural, regional, and organizational boundaries

## Acknowledgments

The authors sincerely thank all the participants in this study who were kind and open enough to share their experiences with us. We would also like to thank all the associations who helped recruit the participants and facilitated the collection of the data for this study.

## Supporting information

**S1 Fig. A conceptual framework for Elderly Daycare Centre. Fig 1 in S1 Fig.**
(PDF)

**S1 Table. Demographic details of Participants. Table 1 in S1 Table.**
(TIF)

## Author contributions

**Conceptualization:** Paralee Opasanunt, Panitsara Leekuan.

**Data curation:** Panitsara Leekuan.

**Formal analysis:** Paralee Opasanunt, Panitsara Leekuan.

**Investigation:** Panitsara Leekuan.

**Methodology:** Paralee Opasanunt, Panitsara Leekuan.

**Project administration:** Panitsara Leekuan.

**Writing – original draft:** Panitsara Leekuan.

**Writing – review & editing:** Paralee Opasanunt, Panitsara Leekuan.

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
