## [Decision Letter · Decision Letter 0]

29 Apr 2025

PONE-D-25-14917Perspectives of Multidisciplinary Healthcare Providers in Elderly Daycare Centres: Challenges, Opportunities and Impacts on Geriatric Care in Chiangrai Municipality, ThailandPLOS ONE

Dear Dr. Leekuan,

Thank you for submitting your manuscript to PLOS ONE. After careful consideration, we feel that it has merit but does not fully meet PLOS ONE’s publication criteria as it currently stands. Therefore, we invite you to submit a revised version of the manuscript that addresses the points raised during the review process.

We look forward to receiving your revised manuscript.

Kind regards,

Le An Pham, Ph.D.,M.D.

Academic Editor

PLOS ONE

Journal Requirements:

2. Please provide additional details regarding participant consent. In the ethics statement in the Methods and online submission information, please ensure that you have specified (1) whether consent was informed and (2) what type you obtained (for instance, written or verbal, and if verbal, how it was documented and witnessed).

3. Please remove all personal information, ensure that the data shared are in accordance with participant consent, and re-upload a fully anonymized data set.

4. We note you have included a table to which you do not refer in the text of your manuscript. Please ensure that you refer to Table 1 in your text; if accepted, production will need this reference to link the reader to the Table.

5. We notice that your supplementary [tables] are included in the manuscript file. Please remove them and upload them with the file type 'Supporting Information'. Please ensure that each Supporting Information file has a legend listed in the manuscript after the references list.

Additional Editor Comments :

Overall Evaluation:

This paper addresses an important and under explored topic in geriatric care by investigating the perspectives of multidisciplinary healthcare providers in elderly daycare centers in Chiangrai Municipality, Thailand. The qualitative design is appropriate, and the findings are clearly structured and well-presented. However, some improvements are necessary to enhance the clarity, rigor, and balance of the manuscript.

Strengths:

The introduction establishes a strong rationale, clearly situating the research within the context of global and national aging trends.

The use of Interpretative Phenomenological Analysis (IPA) and thematic analysis adds depth to the exploration of participant experiences.

The results are rich, supported by participant quotations, and linked meaningfully to existing literature.

Practical implications are thoughtfully drawn, providing valuable insights for policymakers and practitioners.

Areas for Improvement:

Introduction:

The background review, while comprehensive, could be more concise. I suggest focusing more sharply on the identified research gap concerning healthcare provider experiences.

Methods:

Please clarify whether a back-translation procedure was used to validate translated interview data.

Additionally, it would strengthen the methodology to more explicitly acknowledge the limitations of the small sample size and single setting concerning the generalizability of findings.

Results:

Some thematic sections present slightly repetitive participant quotations. I recommend summarizing overlapping responses more succinctly.

Furthermore, the results would benefit from including a wider range of participant views, particularly any divergent or negative experiences, to present a balanced perspective.

Discussion:

While well-integrated with the existing literature, the discussion occasionally overgeneralizes findings. Please ensure interpretations remain grounded within the scope of the study.

I strongly recommend adding a Limitations subsection to candidly discuss sample size, selection bias, setting specificity, and potential translation issues.

Reviewers' comments:

Reviewer's Responses to Questions

**Comments to the Author**

1. Is the manuscript technically sound, and do the data support the conclusions?

Reviewer #1: Yes

Reviewer #2: Yes

2. Has the statistical analysis been performed appropriately and rigorously? 

Reviewer #1: N/A

Reviewer #2: Yes

3. Have the authors made all data underlying the findings in their manuscript fully available?

Reviewer #1: Yes

Reviewer #2: Yes

4. Is the manuscript presented in an intelligible fashion and written in standard English?

Reviewer #1: Yes

Reviewer #2: Yes

5. Review Comments to the Author

Reviewer #1: Thank you for the opportunity to review the manuscript “Perspectives of Multidisciplinary Healthcare Providers in Elderly Daycare Centres: Challenges, Opportunities and Impacts on Geriatric Care in Chiangrai Municipality, Thailand”. The manuscript explores the lived experiences of healthcare providers delivering care within community-based elderly daycare settings. The authors aim to identify key challenges healthcare professionals face, successful practices utilized, and potential opportunities for enhancing geriatric care services through qualitative inquiry. The study used a phenomenological approach to gather data from twelve multidisciplinary providers, including registered nurses, physiotherapists, nutritionists, and public health technical officers, through semi-structured interviews. Data analysis involved thematic and interpretative phenomenological analysis (IPA), revealing four central themes: opportunities in policy implementation, challenging experiences, enhancement of organizational effectiveness, and reflections on program outcomes and impacts. The manuscript highlights essential insights into how multidisciplinary teams perceive and experience geriatric care services, underscoring the significance of integrated, community-oriented care models for older adults. However, numerous critical areas must be addressed to strengthen the manuscript's academic rigor, theoretical coherence, and methodological clarity, as below:

1. The theoretical perspectives are vaguely described and analyzed:

The manuscript does not explicitly articulate a guiding theoretical approach, essential in phenomenological research. As a qualitative methodology, phenomenology seeks to deeply explore individuals' lived experiences to uncover the meanings and essences of phenomena as perceived by the participants. However, the manuscript does not specify whether the phenomenological approach employed aligns with Husserl's descriptive phenomenology, focusing on bracketing researchers' assumptions to reveal the pure essence of experiences, or Heidegger's interpretative phenomenology (hermeneutic phenomenology), which emphasizes the researchers' interpretative engagement with the data and recognizes their preconceived notions as integral to interpretation. The authors are, therefore, encouraged to articulate the theoretical perspectives underpinning their phenomenological research explicitly and then clarify how phenomenology directly shapes data collection procedures, the process of conceptualization, and subsequent interpretation of findings.

In data gathering, specifying the chosen phenomenological approach would better guide and justify the construction of interview questions, the conduct of interviews, and the researchers’ role within these interactions. The authors might benefit from explicitly describing how they managed their assumptions during interviews (through bracketing or interpretive engagement) to enhance transparency and methodological rigor.

Regarding conceptualization, it is recommended that the authors clearly describe the process used to identify and categorize meaningful statements and themes. A detailed explanation of whether descriptive or interpretative phenomenological principles guided this would strengthen the credibility and clarity of theme development.

When interpreting findings, it is advisable to explicitly acknowledge the hermeneutic process, particularly if adopting Heidegger’s interpretative phenomenology. Describing the iterative process of interpretation—the "hermeneutic circle"—would substantially enhance the depth and transparency of the findings. Clearly defining the theoretical approach will significantly improve the manuscript’s theoretical coherence, methodological rigor, and interpretive clarity.

2. The semi-structured interviews are suitable for qualitative phenomenological research, yet the development and validation of the interview guide lack sufficient detail. The authors should clearly describe the process of constructing, piloting, and refining the interview guide, thereby enhancing methodological transparency and ensuring content validity

3. The data analysis approach combining thematic and interpretative phenomenological analysis (IPA) is appropriate; however, the manuscript lacks sufficient procedural details that assure methodological rigor. Detailed accounts of specific practices such as member-checking, peer debriefing, reflexivity strategies, and inter-rater reliability would considerably improve analytical robustness. A clear reflexivity statement addressing the researchers' positionality and potential biases would ensure transparency and enhance credibility.

4. According to the findings, while the identified themes are relevant and informative, they could benefit from deeper theoretical interpretation. The manuscript presents themes descriptively but misses the opportunity to explicitly connect with existing aging theories, stress-coping mechanisms, or organizational behavior theories. Such theoretical integration would enrich the interpretation and increase the manuscript's scholarly value.

5. The implications for practice and policy are implicitly stated but require more precise articulation. The manuscript would benefit significantly from explicitly outlining actionable policy recommendations and practical interventions derived from the findings. Additionally, discussing the potential scalability of identified successful practices across broader contexts would add practical relevance.

6. A comprehensive discussion of limitations, particularly regarding the transferability of findings, would strengthen the manuscript. Explicitly addressing contextual limitations (cultural, regional, and organizational) and suggesting pathways for future research would enhance methodological rigor and academic value.

7. The Introduction section of the manuscript provides comprehensive context and thorough background information on elderly care and daycare centers. However, it is overly detailed and somewhat repetitive, diluting the reader's focus on the study's objectives and significance. I recommend condensing this section by succinctly synthesizing relevant demographic data and policy contexts. Clarify and streamline the narrative by integrating similar ideas, particularly regarding the aging population's implications, healthcare system pressures, and policy frameworks. This conciseness will enhance readability and emphasize the core rationale and objectives of the study.

8. The authors have provided detailed quotes from participants that significantly enrich the manuscript by illustrating the lived experiences of multidisciplinary healthcare providers in elderly daycare centres. However, the presentation of quotes could be optimized for clarity and effectiveness. Specifically, it is recommended that the authors critically evaluate and selectively present quotes to ensure they succinctly reflect key themes and avoid redundancy. The manuscript would benefit from more strategically integrated quotes, emphasizing those that most strongly illustrate the core findings or unique insights relevant to the research objectives. Overly lengthy quotes could be condensed to maintain reader engagement and highlight only the most pertinent data points. The authors will enhance their qualitative findings' overall readability and impact by carefully curating and refining the quoted content.

In conclusion, the manuscript offers meaningful contributions to understanding multidisciplinary geriatric care experiences but requires improvements in theoretical clarity, methodological rigor, and interpretive depth. Addressing these points with MAJOR REVISION would significantly enhance its suitability for publication in PLOS ONE.

Reviewer #2: Paralee and colleagues reported that key aspects of challenges multidisciplinary healthcare providers face and opportunities for enhancement in healthcare delivery systems of the aging population. The method is sound, and the manuscript is clear and well written. I have a few minor comments:

- Please consider providing exclusion criteria when inviting participants through purposive sampling.

- Since your study includes only one physiotherapist and one nutritionist, could coding by profession reveal the identity of the participants? You should consider this carefully.

- I see the title "Pharmacist," in line 378 but it is not described in Demographic details of participants. Is there an error here?

- The description of activities within the care model at the study centers is insufficiently detailed. Could the author provide a rationale for the purposive sampling method employed, specifically addressing the exclusion of physicians?

6. PLOS authors have the option to publish the peer review history of their article (what does this mean?). If published, this will include your full peer review and any attached files.

Reviewer #1: No

Reviewer #2: No

---

## [Author Response · Author response to Decision Letter 1]

21 Jun 2025

All responses were added in Response to reviewers file attached.

---

## [Editor Report · Decision Letter 1]

30 Jun 2025

PONE-D-25-14917R1Perspectives of Multidisciplinary Healthcare Providers in Elderly Daycare Centres: Challenges, Opportunities and Impacts on Geriatric Care in Chiangrai Municipality, ThailandPLOS ONE

Dear Dr. Leekuan,

Thank you for submitting your manuscript to PLOS ONE. After careful consideration, we feel that it has merit but does not fully meet PLOS ONE’s publication criteria as it currently stands. Therefore, we invite you to submit a revised version of the manuscript that addresses the points raised during the review process.

We look forward to receiving your revised manuscript.

Kind regards,

Le An Pham, Ph.D.,M.D.

Academic Editor

PLOS ONE

Journal Requirements:

**Additional Editor Comments:**

Clarify Method Blending: Distinguish IPA from Thematic Analysis or Justify Hybrid Please to confirm in method section

Clarified Theoretical Positioning (Suggested Section: Introduction & Discussion) such as integrated Activity Theory, Person-Centered Care PCC, and Organizational Support Theory OST

ForEx Activity Theory, which posits that active participation in meaningful, socially engaging activities enhances psychological well-being and life satisfaction in older adults. Additionally, Person-Centered Care (PCC) principles guide the analysis by framing care as relational, individualized, and respectful of autonomy. These frameworks are particularly relevant in the context of Elderly Daycare Centres, where physical, emotional, and social care intersect. The lens of Organizational Support Theory (OST) is also applied to interpret how institutional structures shape staff motivation and collaborative effectiveness.

Synthesized participant narratives into conceptual insights in analysis such as Healthcare providers interpreted government mandates not as abstract directives but as dynamic frameworks that guide clinical practice. Weekly planning and debriefing meetings created a cyclical feedback loop, enhancing implementation fidelity. Providers saw themselves as "conduits of policy," translating structural goals into person-centered routines. This aligns with Organizational Support Theory, as structured reflection improved morale and perceived relevance of their work.

Enhance Visual Representation: Include Conceptual Framework or Model

---

## [Author Response · Author response to Decision Letter 2]

14 Aug 2025

1.All references are rechecked in order to complete and correct.

• Reference no. 3 is revised.

• Reference no. 13 is revised

2.This study employed Interpretative Phenomenological Analysis (IPA) as its qualitative methodology in order to explore the lived experiences and meaning-making processes of participant group in depth. Unlike more general approaches to thematic analysis which identify overarching themes across a dataset.

3. Added Theoretical theory: Integrated Activity Theory, Person-Centered Care (PCC) and Organizational Support Theory (OST) in Introduction section

4.Added Theoretical theory: Integrated Activity Theory, Person-Centered Care (PCC) and Organizational Support Theory (OST) in Discussion section

5.Added the organizational structure facilitated robust interprofessional collaboration, which, in turn, boosted motivation

6.Added information about Healthcare providers viewed government commands as adaptable frameworks that informed their daily clinical practice, not as inflexible, theoretical regulations.

7. Added Structured Weekly meetings to support adaptive care delivery

8. Added the core tenets of OST, which support organization values and cares

9. Included Conceptual Framework for Elderly Daycare Centre

---

## [Editor Report · Decision Letter 2]

17 Aug 2025

Perspectives of Multidisciplinary Healthcare Providers in Elderly Daycare Centres: Challenges, Opportunities and Impacts on Geriatric Care in Chiangrai Municipality, Thailand

PONE-D-25-14917R2

Dear Dr. Leekuan,

We’re pleased to inform you that your manuscript has been judged scientifically suitable for publication and will be formally accepted for publication once it meets all outstanding technical requirements.

Kind regards,

Le An Pham, Ph.D.,M.D.

Academic Editor

PLOS ONE
---

## [Editor Report · Acceptance letter]

PONE-D-25-14917R2

PLOS ONE

Dear Dr. Leekuan,

I'm pleased to inform you that your manuscript has been deemed suitable for publication in PLOS ONE. Congratulations! Your manuscript is now being handed over to our production team.

Kind regards,

on behalf of

Professor Le An Pham

Academic Editor

PLOS ONE